# Decoding the JAK-STAT Axis in Colorectal Cancer with AI-HOPE-JAK-STAT: A Conversational Artificial Intelligence Approach to Clinical–Genomic Integration

**DOI:** 10.3390/cancers17142376

**Published:** 2025-07-17

**Authors:** Ei-Wen Yang, Brigette Waldrup, Enrique Velazquez-Villarreal

**Affiliations:** 1PolyAgent, San Francisco, CA 94102, USA; 2Department of Integrative Translational Sciences, Beckman Research Institute of City of Hope, Duarte, CA 91010, USA; 3City of Hope Comprehensive Cancer Center, Duarte, CA 91010, USA

**Keywords:** AI, artificial intelligence, LLM, large language models, cancer, colorectal cancer, cancer genetics, precision oncology, JAK-STAT pathway, AI-agents

## Abstract

Understanding how specific genetic changes affect colorectal cancer (CRC) is essential for improving treatment, especially in younger patients and those receiving chemotherapy. The JAK-STAT signaling pathway plays an important role in cancer development and immune response, but its impact on CRC has been difficult to study due to the complexity of available data. In this study, we developed AI-HOPE-JAK-STAT, a new artificial intelligence platform that allows researchers and clinicians to explore large cancer datasets simply by asking questions in plain English. This tool can quickly analyze how mutations in JAK-STAT genes relate to patient survival, treatment outcomes, and tumor characteristics. Our findings show that some JAK-STAT mutations are linked to better survival in younger patients and in those treated with common chemotherapy. By making complex data more accessible, AI-HOPE-JAK-STAT may accelerate cancer research and help guide more personalized treatment decisions in the future.

## 1. Introduction

Colorectal cancer (CRC) remains one of the most prevalent and lethal malignancies worldwide, ranking third in global cancer incidence and second in cancer-related deaths [1,2,3]. While overall CRC rates have stabilized or declined in older adults, the incidence of early-onset colorectal cancer (EOCRC)—defined as diagnosis before the age of 50—continues to rise across several populations with disproportionate health burdens in the United States [4,5,6,7,8]. This increase has been accompanied by high rates of EOCRC-related mortality [9,10,11].

EOCRC exhibits distinct clinical and molecular features, including a higher frequency of microsatellite instability (MSI), increased tumor mutational burden, immune checkpoint activation, and unique epigenetic signatures [12,13,14,15,16]. These characteristics suggest alternative oncogenic mechanisms that may differ from those driving late-onset CRC (LOCRC), warranting further investigation into pathway-specific alterations across age and ancestry groups.

The Janus kinase-signal transducer and activator of transcription (JAK/STAT) pathway is a major signaling cascade involved in CRC pathogenesis. The JAK/STAT pathway remains underexplored in EOCRC. Aberrant JAK/STAT signaling has been associated with chronic inflammation, tumor immune evasion, and epithelial–mesenchymal transition (EMT), contributing to more aggressive tumor behavior [1,17,18,19,20,21,22] and reduced treatment response in CRC [23,24,25,26,27]. STAT3 activation, in particular, has been linked to stromal invasion and adverse survival outcomes [28,29,30].

Recent studies investigating molecular alterations in JAK/STAT pathway genes among CRC patients have reported higher frequencies of JAK/STAT-related mutations in EOCRC cases among individuals [20,31,32] from populations with different ethnicities [1]. However, no significant differences in JAK/STAT pathway alterations were observed across age or ancestry groups. Survival analyses indicated that JAK/STAT alterations may still be associated with clinical outcomes, although findings were variable across subgroups. These observations highlight the need for tools that can enable the detailed, flexible exploration of molecular data across clinical and genetic contexts.

Advances in artificial intelligence (AI), including the application of large language models (LLMs), are transforming our ability to extract insights from complex biomedical datasets [33,34,35,36,37]. Conversational AI systems can interpret natural language queries and execute analytical workflows in real time, making it easier for researchers and clinicians to interact with genomic and clinical data [38,39,40]. Despite their promise, platforms focused on specific pathway-level interrogation are essential in enabling accurate analysis and integration across multiple data modalities.

To meet this need, we developed AI-HOPE-JAK-STAT, a conversational AI agent specifically designed to investigate JAK/STAT pathway alterations in CRC. The system enables the integration of clinical, genomic, and transcriptomic data through natural language-driven analysis. In this study, we describe the development of AI-HOPE-JAK-STAT, demonstrate its analytical performance by replicating known trends in CRC cohorts, and explore its application in identifying clinically relevant patterns in EOCRC across population groups.

AI-HOPE-JAK-STAT extends our growing suite of conversational AI platforms designed for precision oncology, building on the foundational architectures of AI-HOPE [38], AI-HOPE-TGFbeta [39], and AI-HOPE-PI3K [40]. While previous agents focused on general clinical–genomic queries or specific pathways such as TGF-β and PI3K, AI-HOPE-JAK-STAT uniquely targets the JAK/STAT signaling axis—a critical, yet underexplored, pathway in colorectal cancer. This platform incorporates lessons learned from our prior systems, particularly in optimizing prompt engineering, enhancing reproducibility, and dynamically tailoring analytical workflows in response to user input. Like its predecessors, AI-HOPE-JAK-STAT supports key statistical operations including survival analysis, mutation frequency comparisons, and odds ratio estimation; however, it is distinguished by its ability to stratify analyses by variables such as treatment exposure, microsatellite status, and ancestry. By embedding these functionalities into a natural language-driven interface, AI-HOPE-JAK-STAT empowers users to interrogate the clinical significance of JAK/STAT alterations in real time, facilitating both exploratory hypothesis generation and pathway-specific biomarker discovery.

## 2. Materials and Methods

### 2.1. AI-HOPE-JAK-STAT Platform Design and Functionality

AI-HOPE-JAK-STAT is a custom-built conversational artificial intelligence platform designed to explore the clinical and genomic landscape of JAK/STAT pathway alterations in CRC. The system leverages a LLM-driven interface that translates plain language queries into executable bioinformatics commands (Figure 1). This architecture enables the real-time integration and interrogation of multi-dimensional CRC datasets using a modular backend combining data parsing, statistical modeling, and dynamic visualization.

To ensure reproducibility and accessibility, we have made the full AI-HOPE-JAK-STAT codebase, including natural language-to-code translation logic, data processing modules, and backend bioinformatics workflows, publicly available (see data accessibility statement). The platform uses a LLaMA 3-based large language model deployed locally with predefined prompts optimized for clinical–genomic query translation. It operates via a Python v3.12-based interpreter that converts natural language inputs into executable code, leveraging commonly used packages (e.g., pandas, lifelines, matplotlib) for real-time survival analysis, cohort filtering, and mutation profiling. The system is fully open-source and designed for research use only. This public release ensures that others can reproduce the results and extend the platform for additional use cases in colorectal cancer and beyond.

To provide greater clarity on the system’s technical implementation, AI-HOPE-JAK-STAT is built on a modular Python framework that integrates a locally deployed LLaMA 3-based large language model. The model processes user queries via custom-designed prompts that have been fine-tuned to optimize accuracy in clinical–genomic query translation. The backend uses well-established Python libraries—including pandas for data manipulation, lifelines for survival analysis, matplotlib for visualization, and scipy for statistical testing—to generate and execute code in real time. Each plain language query is parsed through a structured prompt pipeline that extracts key parameters (e.g., gene, age group, treatment exposure) and dynamically constructs the appropriate statistical workflow. The system operates locally for full data privacy and computational control, and its codebase—including prompt logic, data ingestion scripts, statistical modules, and visualization functions—is publicly available (see the data availability statement). This ensures full transparency, reproducibility, and adaptability for users seeking to apply or extend the platform to new datasets or cancer types.

### 2.2. Data Sources and Processing

The platform ingests harmonized CRC datasets derived from the publicly available repository cBioPortal. Data preprocessing involved the normalization of sample identifiers, the removal of incomplete clinical records, and the mapping of genomic features to canonical gene symbols based on HGNC nomenclature. Genes involved in the JAK/STAT signaling pathway—such as JAK1, JAK2, JAK3, TYK2, STAT1, STAT3, STAT5A/B, and SOCS family members—were predefined for targeted analysis. Clinical annotations included tumor stage, anatomical location, treatment history, microsatellite instability (MSI) status, and self-reported ancestry.

### 2.3. Natural Language Input and Query Interpretation

Users interact with the platform through an intuitive natural language interface. AI-HOPE-JAK-STAT processes these inputs using an LLM (LLaMA 3-based) to identify intent, extract parameters, and structure downstream analytic operations. Example queries include the following: “Compare mutation rates in STAT3 between patients under and over 50,” or “Show survival differences in JAK1-mutated tumors treated with chemotherapy.” Ambiguities in user input trigger clarification prompts to maintain analytic precision and reproducibility.

To further clarify the LLM component of AI-HOPE-JAK-STAT, the platform employs a locally deployed LLaMA 3-based model configured with inference parameters optimized for reproducibility and precision: temperature = 0.2, top_*p* = 0.9, and a maximum token length of 2048. These settings were selected to reduce response variability and ensure the consistent translation of natural language inputs into executable code. The system uses structured, domain-specific prompt templates designed to extract analytical intent from user queries. Each prompt includes slots for gene targets, clinical stratifiers (e.g., age, treatment, MSI status), and desired outputs (e.g., Kaplan–Meier plots, contingency tables). The model then translates these prompts into code using predefined syntax rules. Importantly, AI-HOPE-JAK-STAT runs entirely locally, without reliance on third-party APIs, ensuring user data security and analytical reproducibility. If a different LLM is adopted in future versions, the prompt templates may require re-tuning to ensure continued accuracy and relevance. This flexibility has been built into the platform’s modular design to allow for future upgrades.

### 2.4. Analytical Framework

Backend analytics for AI-HOPE-JAK-STAT are implemented in Python, leveraging libraries such as pandas, lifelines, and SciPy v1.11 to execute a range of statistical and bioinformatics operations. The platform performs descriptive summaries to assess mutation frequencies and characterize cohort composition. For categorical comparisons, it employs chi-square or Fisher’s exact tests to evaluate statistical significance. Survival outcomes are analyzed using Kaplan–Meier estimates with log-rank testing, while multivariable survival modeling is conducted using Cox proportional hazards regression. The system also enables flexible subgroup stratification based on age group, self-reported ancestry, treatment exposure, and tumor location. Additionally, AI-HOPE-JAK-STAT supports co-occurrence and mutual exclusivity analyses to explore potential interactions between JAK/STAT pathway genes and other molecular drivers relevant to colorectal tumorigenesis.

### 2.5. Application Validation

To evaluate AI-HOPE-JAK-STAT’s analytical robustness, we applied the platform to replicate findings from previous large-cohort studies investigating JAK/STAT and MAPK pathway alterations across EOCRC and LOCRC populations. Validation tasks included mutation frequency comparisons between different ethnicity EOCRC patients and assessment of survival impact among STAT3-mutated cases.

### 2.6. Output Generation and Interpretation

Following analysis, the system generates comprehensive visual and tabular outputs. These include survival curves, mutation distribution plots, heatmaps, and odds ratio forest plots. Results are accompanied by auto-generated narrative summaries linking observed patterns to published biomedical studies through a retrieval-augmented generation (RAG) module. All outputs are exportable for downstream reporting or figure preparation.

To evaluate the reproducibility of AI-HOPE-JAK-STAT outputs, we conducted repeated analyses using identical natural language queries across multiple runs and computational environments. We confirmed that survival curves, statistical outputs (e.g., *p*-values, odds ratios), and visualizations remained consistent, demonstrating the platform’s reliability in producing stable results. Reproducibility is further supported by version-controlled backend scripts and automated query logging, allowing users to trace and replicate analytical workflows. To enhance the interpretability of the results, the platform also includes a Retrieval-Augmented Generation (RAG) module that generates narrative summaries linking observed findings to relevant biomedical studies. This module retrieves content from a curated index of open-access sources, including selected journals with permissive licensing. Importantly, all RAG-generated summaries presented in this study were manually reviewed and curated by scientific domain experts to ensure contextual relevance, scientific accuracy, and alignment with the underlying data.

### 2.7. Usability and Comparative Evaluation

To assess platform usability, we benchmarked AI-HOPE-JAK-STAT against existing tools (e.g., cBioPortal, Xena Browser) across common analytic tasks such as stratified mutation analysis, co-alteration assessment, and treatment–outcome comparisons. The metrics evaluated included the task completion time, flexibility in cohort definition, and reproducibility of results. AI-HOPE-JAK-STAT demonstrated increased efficiency, especially in generating nested subgroup analyses and complex multi-parameter queries.

## 3. Results

AI-HOPE-JAK-STAT enabled the natural language-driven interrogation of the JAK/STAT signaling cascade across diverse CRC patient populations, integrating genomic alterations with clinical outcomes in real time. Through a series of validation and exploratory analyses, the platform demonstrated its ability to reproduce known associations, uncover novel patterns, and generate interpretable visual and statistical outputs that facilitate hypothesis generation and precision oncology discovery.

To assess ancestry-specific differences in JAK/STAT pathway alterations and their potential clinical significance, we conducted focused analyses using AI-HOPE-JAK-STAT in early-onset colorectal cancer (EOCRC) cohorts stratified by ancestry. As a first example, we queried Hispanic/Latino (H/L) EOCRC patients under 50 years of age and compared overall survival between those harboring JAK/STAT pathway mutations (*n* = 16) and those without such alterations (*n* = 137). In this analysis, presented in Figure 2, the observed *p*-value of 0.0539 does not meet the conventional threshold for statistical significance. Accordingly, we interpret this finding as non-significant and hypothesis-generating, rather than indicative of a definitive survival benefit. While the trend may suggest a potential prognostic signal, it should be interpreted with caution and warrants further validation in larger, independent H/L-specific CRC cohorts. To evaluate the robustness and ancestry-specific reproducibility of this observation, we replicated the analysis in EOCRC patients of Non-Hispanic White (NHW) descent. In this group, patients with JAK/STAT pathway alterations demonstrated a statistically significant survival advantage compared to those without (*p* = 0.0001; Appendix A), reinforcing the potential prognostic value of these mutations. These ancestry-stratified examples illustrate how AI-HOPE-JAK-STAT enables the real-time, hypothesis-driven interrogation of clinical and genomic relationships that would otherwise require extensive manual coding and data wrangling. Although the current version of the tool is designed for research use only, its ability to rapidly contextualize genomic alterations within demographic and clinical subgroups provides a scalable framework for future applications in biomarker discovery and precision oncology.

Next, we investigated treatment context by focusing on CRC patients who received FOLFOX chemotherapy. Kaplan–Meier survival analysis revealed that individuals with JAK/STAT pathway alterations (*n* = 282) had significantly better survival compared to those without such mutations (*n* = 3791) under the same treatment regimen (*p* < 0.0001; Figure 3). These findings suggest that JAK/STAT alterations may be associated with enhanced treatment response in FOLFOX-exposed patients.

To explore age-related effects, we stratified JAK/STAT-altered CRC patients by early- (<50 years; *n* = 133) versus late-onset (≥50 years; *n* = 424) status. Early-onset cases exhibited significantly improved survival (*p* = 0.0379), and odds ratio testing suggested modest enrichment of FOLFOX treatment in this younger group, although this was not statistically significant (OR = 1.356, *p* = 0.154; Figure 4). These results highlight possible interactions between age, treatment exposure, and molecular alterations.

We then assessed the prognostic impact of JAK1 mutations in microsatellite-stable (MSS) tumors. Among MSS CRC patients, those with JAK1 mutations (*n* = 44) showed no significant difference in overall survival relative to wild-type controls (*n* = 4638; *p* = 0.5691; Figure 5). Despite the absence of survival benefit, this analysis underscores the utility of AI-HOPE-JAK-STAT in isolating molecular contexts for precision stratification.

Further, we analyzed STAT5B mutation status in primary tumor samples. Patients with STAT5B-mutated primary tumors (*n* = 109) experienced significantly better overall survival than those without such alterations (*n* = 3945; *p* < 0.0001; as shown in Appendix A). Odds ratio testing revealed that STAT5B-mutated tumors were more frequently located in the colon than in other sites (OR = 1.949, *p* = 0.002), suggesting anatomical preferences in mutation distribution and potential therapeutic relevance.

Finally, we explored stage-specific survival patterns among JAK3-mutated CRC patients. When stratified by disease stage, individuals diagnosed at Stage I–III (*n* = 142) exhibited significantly more favorable survival trends than those with Stage IV tumors (*n* = 36), with highly significant survival differences (*p* < 0.0000; Appendix A). These findings underscore the strong prognostic impact of clinical stage even within molecularly defined subgroups.

Together, these analyses establish AI-HOPE-JAK-STAT as a robust, interactive platform capable of uncovering clinically relevant associations between JAK/STAT signaling alterations and survival outcomes across stratified CRC cohorts. By combining natural language interfaces with integrated bioinformatics, the system supports the reproducible, real-time analysis of complex cancer datasets, offering new pathways toward individualized treatment strategies.

## 4. Discussion

This study presents AI-HOPE-JAK-STAT, a novel conversational artificial intelligence platform designed to interrogate the clinical–genomic landscape of JAK/STAT signaling alterations in CRC. By combining a natural language interface with automated, reproducible bioinformatics workflows, the system addresses longstanding limitations in accessibility and flexibility of pathway-specific data analysis, particularly for EOCRC and other clinically stratified subgroups.

Unlike conventional tools that rely on manual curation or scripting expertise, AI-HOPE-JAK-STAT enables users to conduct complex, multi-layered analyses through natural language queries. Across validation and exploratory tasks, the platform replicated known associations—such as the prognostic impact of JAK/STAT alterations in EOCRC—and uncovered novel trends in treatment response and mutational co-enrichment. In doing so, AI-HOPE-JAK-STAT has demonstrated that conversational AI can serve as a scalable and robust solution for precision oncology research.

Our findings reinforce the biological and clinical relevance of JAK/STAT signaling in CRC. In early-onset patients of both Hispanic/Latino (H/L) and NHW backgrounds, alterations in pathway genes such as JAK1, JAK3, and STAT5B were associated with improved survival, particularly among those receiving FOLFOX chemotherapy. These results align with the preclinical literature suggesting that JAK/STAT dysregulation may sensitize tumors to immune-mediated or cytotoxic interventions. Moreover, the observed stage-specific survival differences among JAK3-mutant tumors (Stage I–III vs. Stage IV) suggest that clinical context significantly modulates the prognostic impact of these alterations.

Beyond outcome comparisons, AI-HOPE-JAK-STAT was also effective in evaluating population-level enrichment. For example, STAT5B mutations were found to be significantly more prevalent in colon versus rectal tumors, an anatomical preference that may reflect divergent etiologies or tissue-specific gene regulation. Conversely, JAK1 mutations within microsatellite-stable (MSS) tumors did not exhibit a survival effect, emphasizing the importance of molecular context in interpreting prognostic significance. Such nuanced insights are rarely obtainable without substantial analytic infrastructure, yet were generated in seconds using AI-HOPE-JAK-STAT’s conversational engine.

A key strength of the platform lies in its adaptability across user-defined comparisons—age, ancestry, tumor stage, and treatment regimen—all seamlessly integrated through structured outputs. This flexibility positions AI-HOPE-JAK-STAT as not just an analytics tool, but a collaborative system for hypothesis testing, education, and translational discovery. The integration of retrieval-augmented generation (RAG) further enhances interpretability by contextualizing results within the existing literature, supporting evidence-based insight generation.

We recognize the importance of reproducibility and contextual reliability in AI-assisted research platforms. In this revision, we strengthened the reproducibility claims of AI-HOPE-JAK-STAT by performing validation tests across multiple executions of the same query, confirming consistency in statistical outputs and visualizations. Moreover, we clarified the role and limitations of the RAG module. While this feature enriches outputs by surfacing the relevant literature, it draws from a curated selection of open-access databases and may not reflect the full breadth of biomedical publications due to indexing or licensing restrictions. As such, the RAG-generated summaries should be interpreted as exploratory aids rather than definitive clinical annotations. These revisions underscore our commitment to transparency, reliability, and the responsible use of AI tools in precision oncology.

We acknowledge that several subgroup analyses presented in this study involve limited sample sizes, which constrain the statistical power and reliability of certain findings. These analyses—such as the 16 early-onset H/L cases in Figure 2 and the 44 JAK1-mutated MSI-Stable patients in Figure 5—are explicitly exploratory in nature and should be interpreted as hypothesis-generating rather than confirmatory. AI-HOPE-JAK-STAT operates as an intelligent conversational system whose analytical capabilities are built upon statistical frameworks validated in our previous work. These prior studies guided the system’s training, the design of prompt-to-code translation, and the integration of standardized statistical methods (e.g., survival analysis via Kaplan–Meier estimates and log-rank testing). Because the platform leverages harmonized, publicly available data from cBioPortal, subgroup availability is inherently constrained by the underlying datasets. Importantly, generating accurate outputs from small, stratified cohorts requires iterative optimization within the system’s inference logic. As such, we emphasize that the analyses presented here are meant to demonstrate the platform’s exploratory and interactive capabilities, and advise against drawing definitive clinical conclusions.

In our analysis of JAK3-mutated colorectal tumors, we observed significantly improved survival among patients diagnosed at Stage I–III compared to those at Stage IV (*p* < 0.00001; Appendix A). While this observation demonstrates the system’s ability to stratify survival outcomes by both mutation status and clinical variables, we acknowledge that the survival difference is most likely driven by stage rather than JAK3 mutation itself serving as an independent prognostic factor. This highlights the importance of incorporating multivariable adjustment to account for confounding variables such as tumor stage, treatment exposure, and MSI status. Future versions of AI-HOPE-JAK-STAT will include built-in support for automated Cox proportional hazards modeling to enable more rigorous, adjusted survival analyses, enhancing the platform’s ability to deliver clinically meaningful and context-aware insights.

The current version of AI-HOPE-JAK-STAT was originally configured and optimized based on the analytical framework used in our prior publication [1], which served as a reference standard for training and validating the system’s performance. The platform integrates predefined statistical logic, subgroup structures, and survival analysis pipelines derived from that work, enabling the dynamic execution of queries using harmonized colorectal cancer datasets from cBioPortal. These foundations ensure that the system generates reproducible and interpretable outputs for hypothesis-driven research. However, while the existing version performs univariate survival analysis (e.g., Kaplan–Meier estimates with log-rank tests), it was not initially configured to automate multivariable modeling. Future developments of AI-HOPE-JAK-STAT will incorporate built-in support for multivariable survival modeling and statistical power estimation, further enhancing the platform’s capability to support robust, exploratory, and translational analyses in precision oncology.

AI-HOPE-JAK-STAT represents the next evolution in our series of conversational AI platforms for precision oncology, offering a pathway-specific framework tailored to interrogate the clinical relevance of JAK/STAT signaling in colorectal cancer. Compared to our earlier agents—AI-HOPE [39], which established the feasibility of natural language-guided multi-omics exploration, and AI-HOPE-TGFbeta [40] and AI-HOPE-PI3K [41], which focused on discrete oncogenic pathways—AI-HOPE-JAK-STAT introduces refined capabilities for handling complex stratification variables such as treatment history, microsatellite status, and self-reported ancestry. This platform also integrates an updated prompt engine, expanded backend analytics, and a reproducibility-focused infrastructure that builds on insights gained from prior iterations. Notably, JAK/STAT pathway alterations present unique analytical challenges due to their diverse roles in immune modulation and tumor progression, and AI-HOPE-JAK-STAT is specifically designed to accommodate these complexities. Together, these advancements demonstrate the growing versatility of our AI-HOPE framework and its potential to support hypothesis-driven, equitable research across biologically distinct signaling pathways in colorectal cancer.

Nevertheless, several limitations warrant discussion. The reliance on publicly available datasets, such as those from cBioPortal, may limit the generalizability of findings due to sample size constraints and potential biases in data curation, particularly the under-representation of certain ancestry groups and uneven clinical annotation across studies. Although AI-HOPE-JAK-STAT supports stratification by ancestry, tumor stage, and treatment, its outputs are inherently shaped by the composition and completeness of the source data. Broader deployment and more equitable insight generation will benefit from the integration of additional multi-omic layers (e.g., RNA-seq [10,41], spatial biology data [42,43,44] and the inclusion of prospective, demographically diverse clinical datasets. Furthermore, while the natural language interface promotes accessibility, performance can vary depending on query specificity. Future iterations may incorporate query refinement tools, dialogue memory, and user feedback loops to enhance interpretability, robustness, and overall user experience.

Despite these limitations, the platform’s performance in recapitulating prior findings, identifying novel associations, and supporting pathway-level analyses without coding expertise reflects a meaningful advancement in AI-enabled cancer research. AI-HOPE-JAK-STAT not only complements traditional tools like cBioPortal and Xena but offers unique advantages in speed, flexibility, and hypothesis exploration.

## 5. Conclusions

AI-HOPE-JAK-STAT offers a promising addition to the evolving precision oncology landscape by enabling the natural language-guided integration of genomic, clinical, and treatment data for the investigation of JAK/STAT pathway alterations in colorectal cancer (CRC). The platform facilitates real-time, hypothesis-driven analyses—such as survival trends, mutation enrichment, and ancestry- or treatment-specific subgroup exploration—without requiring programming expertise. These features may be particularly useful in examining the heterogeneity of early-onset CRC and generating insights that could support the design of future clinical studies and personalized treatment strategies. While AI-HOPE-JAK-STAT demonstrates the feasibility of conversational AI for pathway-level analysis, its broader utility will depend on continued development, the integration of multivariable modeling, and validation across more diverse, prospective datasets. As precision oncology moves toward more individualized and immunologically informed care, tools like AI-HOPE-JAK-STAT may help support biomarker discovery and enhance accessibility to integrative data analysis across varied research and clinical environments.

## Figures and Tables

**Figure 1 cancers-17-02376-f001:**
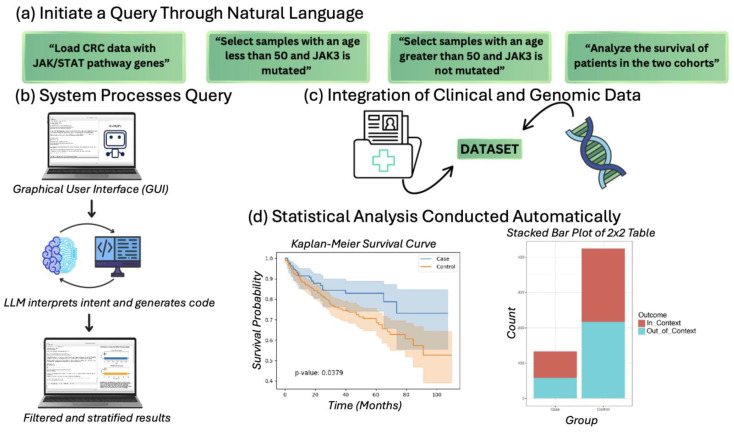
Workflow of AI-HOPE-JAK-STAT for clinical–genomic analysis of the JAK/STAT pathway in colorectal cancer. This figure illustrates the operational framework of AI-HOPE-JAK-STAT, a conversational artificial intelligence system designed to explore JAK/STAT signaling alterations in colorectal cancer (CRC) using natural language input. (**a**) Users initiate analysis by entering plain language queries such as loading CRC datasets with JAK/STAT pathway genes, filtering by age or mutation status (e.g., JAK3), and requesting survival comparisons between defined cohorts. (**b**) The system interprets the query through a graphical user interface (GUI) supported by a large language model (LLM), which parses the user’s intent, generates code, and executes the relevant operations. (**c**) Clinical and genomic data are automatically retrieved and integrated from harmonized datasets. These include patient-level variables and mutation profiles relevant to the JAK/STAT signaling axis. (**d**) Statistical analyses are then conducted automatically, producing visual outputs such as Kaplan–Meier survival curves and contingency plots. Results are stratified according to user-defined parameters and delivered alongside narrative interpretations, enabling the streamlined investigation of JAK/STAT-driven tumor biology in CRC.

**Figure 2 cancers-17-02376-f002:**
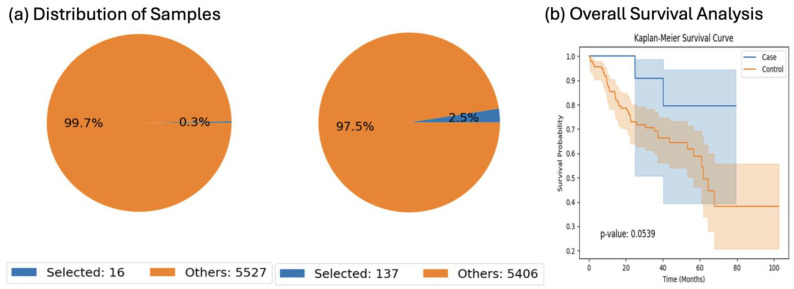
AI-HOPE-JAK-STAT analysis of early-onset colorectal cancer patients of ethnicity-specific descent stratified by JAK/STAT pathway alteration status. This figure presents the output of a natural language query executed via AI-HOPE-JAK-STAT to investigate the impact of JAK/STAT pathway alterations on overall survival in early-onset colorectal cancer (EOCRC) patients of Hispanic/Latino descent (H/L). The case cohort includes patients under age 50 with alterations in JAK/STAT pathway genes, while the control cohort includes similarly aged patients of the same ancestry group without such alterations. (**a**) Pie charts display the proportional representation of selected samples in each cohort relative to the full dataset (*n* = 5543), highlighting the rarity of JAK/STAT pathway alterations in this subgroup. (**b**) Kaplan–Meier survival analysis shows a trend toward improved survival in the JAK/STAT-altered cohort (blue) compared to the non-altered group (orange). Although the *p*-value of 0.0539 from the log-rank test does not reach conventional statistical significance (*p* < 0.05), the observed difference suggests a possible survival benefit that may warrant further validation in larger cohorts. Shaded confidence intervals illustrate variability in the estimates and highlight the importance of cautious interpretation.

**Figure 3 cancers-17-02376-f003:**
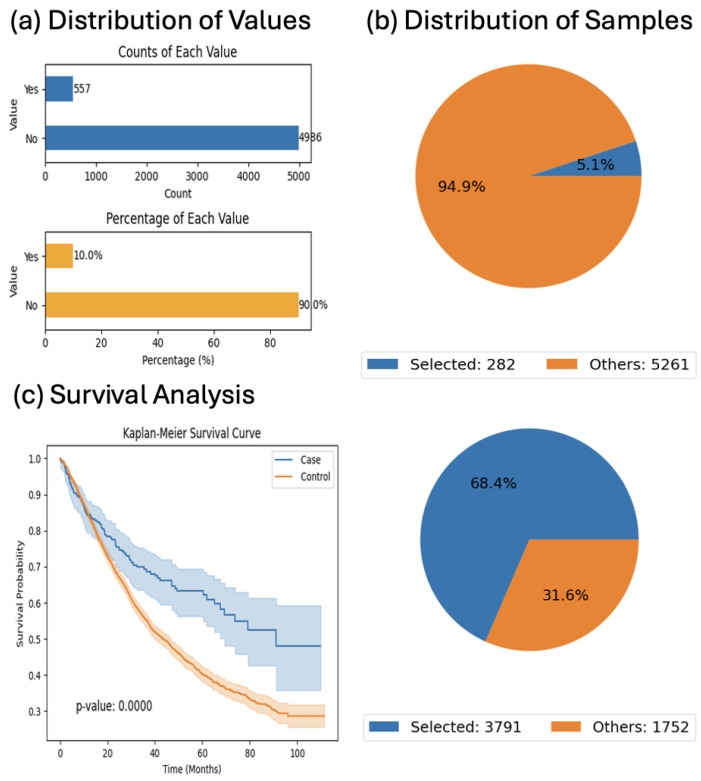
AI-HOPE-JAK-STAT analysis of colorectal cancer patients treated with FOLFOX stratified by JAK/STAT pathway alteration status. This figure presents the results of a natural language-driven analysis conducted through AI-HOPE-JAK-STAT, evaluating overall survival among colorectal cancer (CRC) patients treated with combination chemotherapy (Fluorouracil, Leucovorin, and Oxaliplatin—FOLFOX), stratified by JAK/STAT pathway alteration status. The case cohort includes patients with JAK/STAT alterations who received FOLFOX (*n* = 282), while the control cohort includes patients without such alterations who received the same regimen (*n* = 3791). (**a**) Bar plots summarize the distribution of JAK/STAT pathway alterations across the entire dataset (*n* = 5543), showing that 10% of patients exhibit alterations in this pathway. The top chart displays absolute counts, while the bottom chart presents percentages. (**b**) Pie charts illustrate the relative representation of selected patients in each cohort. The case cohort accounts for 5.1% of the dataset, while the control cohort constitutes 68.4%, reflecting the larger proportion of patients without JAK/STAT alterations receiving FOLFOX. (**c**) Kaplan–Meier survival analysis reveals a statistically significant survival advantage in the JAK/STAT-altered cohort (blue) compared to the control group (orange), with a *p*-value < 0.0001. The clear separation between curves and narrow confidence intervals suggest a robust association between JAK/STAT pathway alterations and improved treatment outcomes among CRC patients undergoing FOLFOX chemotherapy.

**Figure 4 cancers-17-02376-f004:**
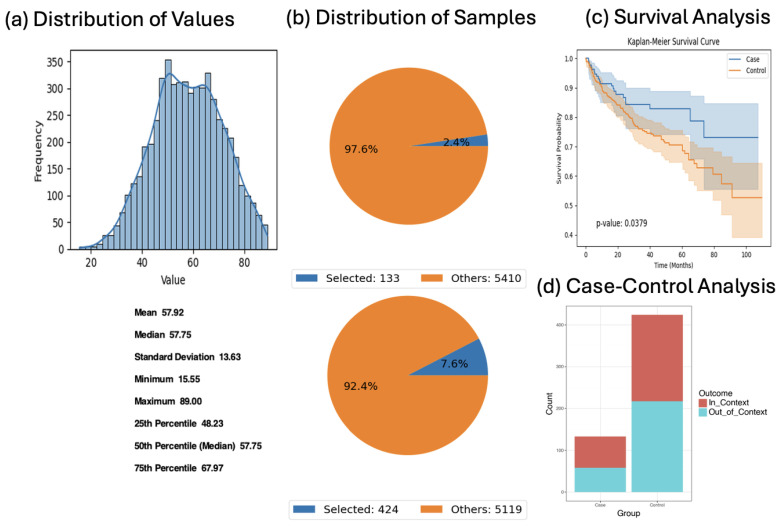
AI-HOPE-JAK-STAT analysis of age-stratified colorectal cancer patients with JAK/STAT pathway alterations treated with FOLFOX. This figure summarizes the results of a natural language query executed via AI-HOPE-JAK-STAT to compare survival outcomes and chemotherapy context between colorectal cancer (CRC) patients with JAK/STAT pathway alterations, stratified by age. The case cohort consists of patients younger than 50 years (*n* = 133), and the control cohort includes patients older than 50 (*n* = 424). All individuals received combination chemotherapy with Fluorouracil, Leucovorin, and Oxaliplatin (FOLFOX). (**a**) A histogram displays the age distribution of the full dataset (*n* = 5543), with a mean age of 57.92 years, confirming the rationale for early- vs. late-onset cohort definitions. (**b**) Pie charts visualize the proportion of selected samples within each cohort relative to the entire population. The early-onset (case) group comprises 2.4% of samples, while the late-onset (control) group represents 7.6%. (**c**) Kaplan–Meier survival analysis reveals a statistically significant survival difference between the two age groups, with the early-onset JAK/STAT-altered cohort (blue) demonstrating superior outcomes compared to the older group (orange) (*p* = 0.0379). (**d**) A 2 × 2 odds ratio analysis was conducted to assess treatment context by comparing the number of in-context and out-of-context samples (based on receipt of FOLFOX chemotherapy) across cohorts. The stacked bar plot shows that 56.39% of early-onset cases and 48.82% of late-onset controls were in the user-defined treatment context. The resulting odds ratio was 1.356 (95% CI: 0.916–2.006, *p* = 0.154), suggesting a modest but non-significant enrichment of FOLFOX treatment in the early-onset group.

**Figure 5 cancers-17-02376-f005:**
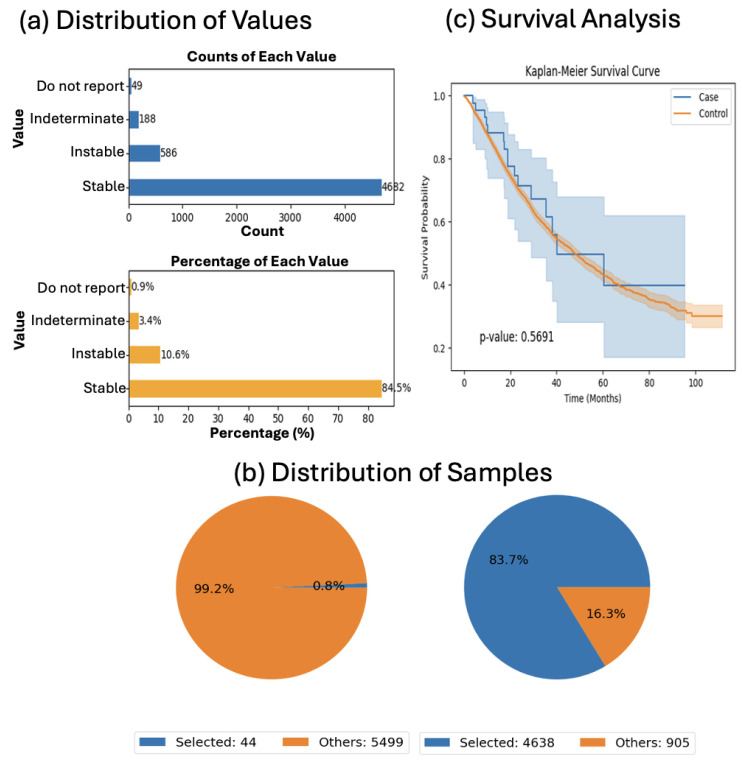
AI-HOPE-JAK-STAT analysis of JAK1 mutation status in microsatellite-stable colorectal cancer. This figure illustrates the use of AI-HOPE-JAK-STAT to investigate overall survival differences in colorectal cancer (CRC) patients with stable microsatellite status (MSI-Stable), stratified by JAK1 mutation status. The case cohort includes MSI-Stable patients harboring JAK1 mutations (*n* = 44), while the control cohort includes MSI-Stable patients without JAK1 mutations (*n* = 4638). (**a**) Bar plots display the distribution of MSI types across the full dataset using simplified labels (e.g., X0 = Stable, X1 = Instable). MSI-Stable tumors represent the majority of samples, confirming that the data are sufficient to evaluate JAK1-specific outcomes within this molecular context. (**b**) Pie charts show the relative proportion of selected samples in the case and control cohorts. The JAK1-mutant group comprises only 0.8% of the dataset, compared to 83.7% for the JAK1 wild-type group within MSI-Stable cases, underscoring the rarity of this alteration. (**c**) Kaplan–Meier survival analysis reveals no statistically significant difference in overall survival between the two cohorts (*p* = 0.5691). Although the JAK1-mutant group shows a slightly different survival trajectory, the wide confidence intervals and overlapping curves indicate no clear association with survival in this subset. These results suggest that JAK1 mutation status may not independently influence prognosis among MSI-Stable CRC patients and highlight the importance of considering broader molecular contexts in precision oncology studies.

## Data Availability

The data used in this study is available to the public and can be found at cbioportal.org. The AI-HOPE-JAK-STAT software, along with demonstration data and documentation, is available at https://github.com/Velazquez-Villarreal-Lab/AI-JAK-STAT (accessed on 11 July 2025).

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
