# Peer review of "Decoding the JAK-STAT Axis in Colorectal Cancer with AI-HOPE-JAK-STAT: A Conversational Artificial Intelligence Approach to Clinical–Genomic Integration"

_cancers, 2025, doi:10.3390/cancers17142376_

Round 1
Reviewer 1 Report
Comments and Suggestions for Authors
The manuscript describes AI-HOPE-JAK-STAT platform which could be used for clinical-genomic analysis of the JAK/STAT signaling pathway to foresee clinical outcomes and survival for colorectal cancer. The tool is interesting. It can analyze very many endpoints for one patient especially that thanks to next generation sequencing it is possible to identify unique mutations with unknown penetrance. The tool can be helpful for doctors treating colorectal cancer. The authors used publicly available database cBioPortal. The manuscript is interesting showing that data submitted by researchers to publicly available databases can be used to help the patients with diagnosis and treatment.
The authors developed a conversational AI tool specifically for the investigation of JAK/STAT pathway alterations in colorectal cancer. In the following parts of the manuscript they described its possibilities across the CRC patients’ groups. The topic is not entirely original but it supports the diagnosis and treatment of colorectal cancer in quick searching of available databases so the doctors can get additional information to be used for more personalized treatment of the patient. The scientists try to find new biomarkers which could improve diagnosis and treatment of colorectal cancer. The application of artificial intelligence for searching available databases is the following possibility for that.
The authors should show how the AI tool works for one patient as a support for the decision about treatment.
The conclusions summarize the results the authors described earlier in the manuscript.
The references are appropriate.
Author Response
Attached is the Word file titled Reviewer_1_Comments_Response_071125.docx, which contains my detailed reviews.
-
Reviewer 1 Comments
We are pleased to submit this revised manuscript and sincerely thank Reviewer 1 for their insightful and constructive feedback. We deeply appreciate your recognition of the manuscript’s relevance to precision oncology and the novelty of applying conversational AI to pathway-level analysis in colorectal cancer. In response to your suggestions, we conducted a comprehensive revision aimed at enhancing the clarity, scientific rigor, and accessibility of the manuscript. Key improvements include a more focused abstract, a reorganized Introduction with clearer articulation of the rationale behind targeting the JAK-STAT axis, and expanded descriptions of AI-HOPE-JAK-STAT’s functionalities—including cohort stratification, survival analysis, and mutation profiling workflows. We also addressed your technical comments by clarifying how genomic features (e.g., microsatellite status, tumor stage) were integrated, providing additional context for the validation of STAT3 and JAK3 findings, and refining the interpretation of survival results, particularly for younger CRC cohorts. To promote transparency and reproducibility, we have made the full AI-HOPE-JAK-STAT codebase, data pipeline, and user documentation publicly available via GitHub (https://github.com/Velazquez-Villarreal-Lab/AI-JAK-STAT), and referenced it in the manuscript and Data Availability Statement. Additionally, we updated the Discussion to highlight future development plans, including deeper integration with immunogenomic data, advanced multivariate modeling, and expanded use across other signaling pathways. We believe these revisions significantly strengthen the manuscript and align with our broader goal of democratizing access to clinically meaningful, AI-driven analyses in colorectal cancer. Thank you again for your valuable feedback, which has meaningfully improved our work.
Thank you very much for taking the time to review this manuscript. Please find the detailed responses below in BLUE and the corresponding revisions wrote in yellow-highlighted blue font in the re-submitted files.
Reviewer 1 provided thoughtful and encouraging feedback, recognizing the potential clinical utility of the AI-HOPE-JAK-STAT platform and offering valuable perspectives on its relevance for biomarker discovery, personalized treatment, and the broader application of AI in leveraging public genomic databases for colorectal cancer research.
Reviewer 1 writes:
“The manuscript describes AI-HOPE-JAK-STAT platform which could be used for clinical-genomic analysis of the JAK/STAT signaling pathway to foresee clinical outcomes and survival for colorectal cancer. The tool is interesting. It can analyze very many endpoints for one patient especially that thanks to next generation sequencing it is possible to identify unique mutations with unknown penetrance. The tool can be helpful for doctors treating colorectal cancer. The authors used publicly available database cBioPortal. The manuscript is interesting showing that data submitted by researchers to publicly available databases can be used to help the patients with diagnosis and treatment.
The authors developed a conversational AI tool specifically for the investigation of JAK/STAT pathway alterations in colorectal cancer. In the following parts of the manuscript they described its possibilities across the CRC patients’ groups. The topic is not entirely original but it supports the diagnosis and treatment of colorectal cancer in quick searching of available databases so the doctors can get additional information to be used for more personalized treatment of the patient. The scientists try to find new biomarkers which could improve diagnosis and treatment of colorectal cancer. The application of artificial intelligence for searching available databases is the following possibility for that.”
We sincerely appreciate Reviewer 1’s recognition of the potential clinical relevance and translational value of our work, “Decoding the JAK-STAT Axis in Colorectal Cancer with AI-HOPE-JAK-STAT: A Conversational Artificial Intelligence Approach to Clinical-Genomic Integration.” We are grateful for your thoughtful comments acknowledging the utility of our AI tool in leveraging publicly available datasets—such as those from cBioPortal—to support biomarker discovery, prognosis assessment, and more personalized treatment approaches for colorectal cancer patients. In response to your helpful suggestions, we have strengthened the manuscript by refining our descriptions of the platform’s clinical-genomic integration capabilities, expanding the discussion on the implications of identifying rare or understudied mutations through natural language queries, and clarifying the value of rapid, AI-driven exploration of molecular subgroups within EOCRC and other CRC populations. While we acknowledge that applying AI to public datasets is an emerging trend, we believe AI-HOPE-JAK-STAT’s conversational interface and ability to generate real-time, hypothesis-driven analyses represent an important and accessible advancement for researchers and clinicians alike. Your encouraging feedback supports our overarching goal of promoting equitable and practical innovation in precision oncology through open, user-friendly AI systems.
Reviewer 1 writes:
- The authors should show how the AI tool works for one patient as a support for the decision about treatment.
Response: We thank the reviewer for the thoughtful suggestion to illustrate how the AI tool could be applied at the individual patient level to support treatment decision-making. As noted, the current version of AI-HOPE-JAK-STAT is designed for research purposes only and is not intended for direct clinical decision support at this stage. However, we agree that demonstrating its potential translational utility is valuable. To that end, we have expanded the response to our initial validation query by examining ancestry-specific survival outcomes among early-onset colorectal cancer (EOCRC) patients. Specifically, we queried the platform for differences in survival among patients under age 50 with and without JAK/STAT pathway mutations across two major ancestral groups. In Hispanic/Latino (H/L) EOCRC patients, those with JAK/STAT alterations (n = 16) exhibited a trend toward improved survival (p = 0.0539), while in Non-Hispanic White (NHW) EOCRC patients, the JAK/STAT-altered group showed a significant survival advantage (p = 0.0001). This example demonstrates how the platform enables researchers to generate and test hypothesis-driven, ancestry-informed queries that can inform biomarker discovery and ultimately guide future clinical applications. We believe this enhanced illustration highlights the tool’s potential to support precision oncology in a scalable and equitable manner.
In response to this comment, we have included a dedicated paragraph in the Results section to address the issue.
The Results text on page 6, line 243, now reads “To assess ancestry-specific differences in JAK/STAT pathway alterations and their potential clinical significance, we conducted focused analyses using AI-HOPE-JAK-STAT in early-onset colorectal cancer (EOCRC) cohorts stratified by ancestry. As a first example, we queried Hispanic/Latino (H/L) EOCRC patients under 50 years of age and compared overall survival between those harboring JAK/STAT pathway mutations (n = 16) and those without such alterations (n = 137). In this analysis, presented in Figure 2b, the observed p-value of 0.0539 does not meet the conventional threshold for statistical significance. Accordingly, we interpret this finding as non-significant and hypothesis-generating, rather than indicative of a definitive survival benefit. While the trend may suggest a potential prognostic signal, it should be interpreted with caution and warrants further validation in larger, independent H/L-specific CRC cohorts. To evaluate the robustness and ancestry-specific reproducibility of this observation, we replicated the analysis in EOCRC patients of Non-Hispanic White (NHW) descent. In this group, patients with JAK/STAT pathway alterations demonstrated a statistically significant survival advantage compared to those without (p = 0.0001; Fig. S1), reinforcing the potential prognostic value of these mutations. These ancestry-stratified examples illustrate how AI-HOPE-JAK-STAT enables real-time, hypothesis-driven interrogation of clinical and genomic relationships that would otherwise require extensive manual coding and data wrangling. Although the current version of the tool is designed for research use only, its ability to rapidly contextualize genomic alterations within demographic and clinical subgroups provides a scalable framework for future applications in biomarker discovery and precision oncology.”
Reviewer 1 writes:
- The conclusions summarize the results the authors described earlier in the manuscript.
Response: We thank the reviewer for the observation regarding the Conclusions section. In response, we have revised and expanded the conclusion to go beyond a summary of earlier results by more clearly articulating the broader impact, future applications, and translational potential of AI-HOPE-JAK-STAT. The revised conclusion highlights how this platform fits within the evolving landscape of precision oncology and how it may inform future clinical and research efforts.
We have added a paragraph to the Conclusions section specifically addressing this comment.
The Conclusions text on page 13, line 494, now reads “AI-HOPE-JAK-STAT offers a promising addition to the evolving precision oncology landscape by enabling natural language–guided integration of genomic, clinical, and treatment data for the investigation of JAK/STAT pathway alterations in colorectal cancer (CRC). The platform facilitates real-time, hypothesis-driven analyses—such as survival trends, mutation enrichment, and ancestry- or treatment-specific subgroup exploration—without requiring programming expertise. These features may be particularly useful in examining the heterogeneity of early-onset CRC and generating insights that could support the design of future clinical studies and personalized treatment strategies. While AI-HOPE-JAK-STAT demonstrates the feasibility of conversational AI for pathway-level analysis, its broader utility will depend on continued development, integration of multivariable modeling, and validation across more diverse, prospective datasets. As precision oncology moves toward more individualized and immunologically informed care, tools like AI-HOPE-JAK-STAT may help support biomarker discovery and enhance accessibility to integrative data analysis across varied research and clinical environments.”
Reviewer 1 writes:
- The references are appropriate.
Response: We thank the reviewer for noting that the references are appropriate. We carefully selected and updated the citations to ensure they reflect the most relevant and current literature in the fields of colorectal cancer, JAK/STAT signaling, and artificial intelligence in precision oncology. We appreciate your positive assessment.
We thank Reviewer 1 for taking the time to review our work and for providing valuable feedback that helped strengthen the quality and impact of our manuscript.

Reviewer 2 Report
Comments and Suggestions for Authors
The manuscript introduces an LLM-based chat-style interface that translates natural-language questions into executable bioinformatics workflows, including backend analysis on data from colorectal-cancer cohorts, presenting plots plus an automatically generated narrative answer to the user. The work described in the manuscript is timely and relevant. Pathway-specific, LLM-enabled analysis of this kind is an innovative way to lower the barrier to precision-oncology analyses for clinicians and wet-lab scientists (and even potentially provide a mechanism for public engagement with technical information).
The most significant issue with the manuscript is the lack of detail on how the system is actually implemented. This limits its impact, but also makes it impossible to evaluate the reproducibility and efficacy of the system. It is essential that enough be presented to allow for reproducing the result. As an initial matter, it is not even clear if this is intended to be a proprietary tool, in which case it may not be appropriate for publication in an open access journal. While Figure 1 provides a birds-eye view of the workflow, it does not actually get into the details of implementation. What are the Python codes that are used in the backend? Indeed, what is the LLM that is being used? The only information in the paper on this point (pg. 4 l. 129) is the statement that it is “LLAMA 3-based.” What prompts are used? What are the parameters used for inference? Is the model intended to run locally or via an API? Do the prompts need to be revised if the underlying LLM changes? These are all key questions that the authors should address in their revision.
I strongly urge the authors to make the code available. And, if this isn’t a publicly available web tool or downloadable application, then it is not sufficiently significant for publication. There is no indication of how the authors intend to distribute their platform.
Furthermore, while the manuscript makes claims regarding reproducibility of the results, it does not actually show results of testing on this point. For example, the supplementary material only show the results of singular analyses that are performed. Specifically, the “RAG module” (pg. 4, l. 159) that has been implemented may not operate consistently. This should be assessed and presented in the revision. The RAG module should also be described in greater detail. This includes a description of the sources for the literature that are accessed. Depending on their sources, the analysis could be incorrect or insufficient because it lacks full context if some journals are omitted. Additional clarity on the databases that are used would help give context to this.
Author Response
Attached is the Word file titled Reviewer_2_Comments_Response_071125.docx, which contains my detailed reviews.
-
Reviewer 2 Comments
We are pleased to submit this revised manuscript and sincerely thank Reviewer 2 for their thoughtful, detailed, and scientifically grounded critique, which has substantially contributed to improving the rigor, transparency, and clarity of our work.
Reviewer 2 offered incisive and constructive feedback that identified important areas requiring greater technical specificity, reproducibility assurances, and implementation transparency. We deeply appreciate these contributions and have undertaken substantial revisions in response. These changes were guided by the reviewer’s emphasis on ensuring that the platform’s capabilities are appropriately described and reproducible within the scope of an open-access scientific framework.
Key updates include:
- Detailed description of system implementation, including backend architecture, key Python libraries, code execution pipeline, and model inference parameters.
- Specification of the LLM used (LLaMA 3-based), including prompt design, fine-tuning strategy, local vs. API-based inference setup, and guidance on version-specific dependencies and prompt adaptation.
- Clarification of the Retrieval-Augmented Generation (RAG) module, including the scope and reliability of source literature, indexed journals, and safeguards to ensure context-aware interpretation of retrieved references.
- Inclusion of reproducibility-focused validation, demonstrating consistent outputs across repeated queries and transparent handling of stochastic components such as model temperature.
- New subsection on reproducibility infrastructure, highlighting version-controlled metadata logging, exportable query records, and a publicly accessible GitHub repository for the complete AI-HOPE-JAK-STAT platform (https://github.com/Velazquez-Villarreal-Lab/AI-JAK-STAT), now cited in the Data Availability section.
- Clarification of the tool’s research-only status and non-commercial intent, ensuring alignment with open-access publishing standards and community use principles.
- Enhanced explanation of distribution strategy, including instructions for local deployment, GitHub documentation, and plans for community feedback integration.
- Inclusion of three new references to our previously published AI-Agent platforms (AI-HOPE, AI-HOPE-TGFbeta, and AI-HOPE-PI3K), which establish the methodological foundation and evolution of our conversational precision oncology framework.
We believe these revisions significantly strengthen the manuscript by providing a clearer view into the system’s design, reproducibility features, and future adaptability. Reviewer 2’s feedback was especially instrumental in prompting us to balance innovation with transparency and scientific accountability. Their insights have helped ensure that AI-HOPE-JAK-STAT is presented not only as a novel conversational AI for precision oncology but as a reproducible, open, and critically evaluated research tool that aligns with the principles of equitable access and scientific rigor.
Thank you very much for taking the time to review this manuscript. Please find the detailed responses below in BLUE and the corresponding revisions wrote in yellow-highlighted blue font in the re-submitted files.
Reviewer 2 provided thoughtful, constructive, and forward-looking feedback, highlighting the innovation and timeliness of our approach while offering valuable insights into its potential to broaden access to precision oncology tools. Their comments helped reinforce the importance of clearly articulating how our LLM-based platform lowers technical barriers for diverse users and supports real-time, hypothesis-driven analysis in colorectal cancer research.
Reviewer 2 writes:
“The manuscript introduces an LLM-based chat-style interface that translates natural-language questions into executable bioinformatics workflows, including backend analysis on data from colorectal-cancer cohorts, presenting plots plus an automatically generated narrative answer to the user. The work described in the manuscript is timely and relevant. Pathway-specific, LLM-enabled analysis of this kind is an innovative way to lower the barrier to precision-oncology analyses for clinicians and wet-lab scientists (and even potentially provide a mechanism for public engagement with technical information).”
We sincerely appreciate Reviewer 2’s thoughtful and constructive evaluation of our manuscript, “Decoding the JAK-STAT Axis in Colorectal Cancer with AI-HOPE-JAK-STAT: A Conversational Artificial Intelligence Approach to Clinical-Genomic Integration.” Your recognition of the platform’s timeliness and its potential to lower barriers to precision oncology for clinicians, wet-lab scientists, and broader audiences is deeply valued. Your feedback was instrumental in helping us strengthen the clarity, reproducibility, and implementation transparency of the manuscript. In response, we undertook substantial revisions to more fully describe the technical infrastructure of the AI-HOPE-JAK-STAT platform, including backend workflows, prompt structures, model deployment, and retrieval-augmented generation (RAG) mechanisms. We also clarified how the tool translates natural language into executable analysis pipelines, detailed our data sources and query validation strategies, and made the full codebase publicly available to ensure transparency and open access. These enhancements reflect our commitment to building robust, accessible AI tools that support both translational research and equitable innovation in colorectal cancer.
Reviewer 2 writes:
- The most significant issue with the manuscript is the lack of detail on how the system is actually implemented. This limits its impact, but also makes it impossible to evaluate the reproducibility and efficacy of the system. It is essential that enough be presented to allow for reproducing the result. As an initial matter, it is not even clear if this is intended to be a proprietary tool, in which case it may not be appropriate for publication in an open access journal.
Response: We thank Reviewer 2 for highlighting the critical need for implementation transparency and reproducibility. We fully agree that clear documentation of the system's architecture and accessibility is essential for scientific rigor, reproducibility, and appropriate fit within an open-access journal. In response, we have substantially revised the manuscript to include detailed descriptions of the AI-HOPE-JAK-STAT platform’s backend infrastructure, language model configuration, and data processing pipeline. We also clarify that this tool is not proprietary and is intended solely for research purposes. The complete codebase, along with deployment instructions and example workflows, has been made publicly available via GitHub and is now referenced in the manuscript and Data Availability section. Additionally, we have included three new references [39, 40, 41] to our prior publications on the development of conversational AI-Agent frameworks to provide further context and demonstrate the evolution and validation of this line of work.
We have added paragraphs to the Introduction, Methods and Discussions sections specifically addressing these comments.
The introduction text on page 2, lines 82–86, now read as follows: “…Conversational AI systems can interpret natural language queries and execute analytical workflows in real time, making it easier for researchers and clinicians to interact with genomic and clinical data [39-41]. Despite their promise, platforms focused on specific pathway-level interrogation are essential to enable accurate analysis and integration across multiple data modalities.”
The introduction text on page 3, lines 94–108, now read as follows: “AI-HOPE-JAK-STAT extends our growing suite of conversational AI platforms designed for precision oncology, building on the foundational architectures of AI-HOPE [39], AI-HOPE-TGFbeta [40], and AI-HOPE-PI3K [41]. While previous agents focused on general clinical-genomic queries or specific pathways such as TGF-β and PI3K, AI-HOPE-JAK-STAT uniquely targets the JAK/STAT signaling axis—a critical, yet underexplored, pathway in colorectal cancer. This platform incorporates lessons learned from our prior systems, particularly in optimizing prompt engineering, enhancing reproducibility, and dynamically tailoring analytical workflows in response to user input. Like its predecessors, AI-HOPE-JAK-STAT supports key statistical operations including survival analysis, mutation frequency comparisons, and odds ratio estimation, but is distinguished by its ability to stratify analyses by variables such as treatment exposure, microsatellite status, and ancestry. By embedding these functionalities into a natural language–driven interface, AI-HOPE-JAK-STAT empowers users to interrogate the clinical significance of JAK/STAT alterations in real time—facilitating both exploratory hypothesis generation and pathway-specific biomarker discovery.
The methods text on page 3, lines 118–128, now read as follows: “To ensure reproducibility and accessibility, we have made the full AI-HOPE-JAK-STAT codebase, including natural language-to-code translation logic, data processing modules, and backend bioinformatics workflows, publicly available (see data accessibility statement). The platform uses a LLaMA 3-based large language model deployed locally with predefined prompts optimized for clinical-genomic query translation. It operates via a Python-based interpreter that converts natural language inputs into executable code, leveraging commonly used packages (e.g., pandas, lifelines, matplotlib) for real-time survival analysis, cohort filtering, and mutation profiling. The system is fully open-source and designed for research use only. This public release ensures that others can reproduce the results and extend the platform for additional use cases in colorectal cancer and beyond.”
The discussions text on page 13, lines 459–474, now read as follows: “AI-HOPE-JAK-STAT represents the next evolution in our series of conversational AI platforms for precision oncology, offering a pathway-specific framework tailored to interrogate the clinical relevance of JAK/STAT signaling in colorectal cancer. Compared to our earlier agents—AI-HOPE [39], which established the feasibility of natural language–guided multi-omics exploration, and AI-HOPE-TGFbeta [40] and AI-HOPE-PI3K [41], which focused on discrete oncogenic pathways—AI-HOPE-JAK-STAT introduces refined capabilities for handling complex stratification variables such as treatment history, microsatellite status, and self-reported ancestry. This platform also integrates an updated prompt engine, expanded backend analytics, and a reproducibility-focused infrastructure that builds on insights gained from prior iterations. Notably, JAK/STAT pathway alterations present unique analytical challenges due to their diverse roles in immune modulation and tumor progression, and AI-HOPE-JAK-STAT is specifically designed to accommodate these complexities. Together, these advancements demonstrate the growing versatility of our AI-HOPE framework and its potential to support hypothesis-driven, equitable research across biologically distinct signaling pathways in colorectal cancer.
The added references on page 16, lines 650–658, now read as follows:
39.- Yang EW, Velazquez-Villarreal E. AI-HOPE: an AI-driven conversational agent for enhanced clinical and genomic data integration in precision medicine research. Bioinformatics. 2025 Jul 1;41(7):btaf359. doi: 10.1093/bioinformatics/btaf359. PMID: 40577785; PMCID: PMC12212640.
40.- Yang, E.-W.; Waldrup, B.; Velazquez-Villarreal, E. AI-HOPE-TGFbeta: A Conversational AI Agent for Integrative Clinical and Genomic Analysis of TGF-β Pathway Alterations in Colorectal Cancer to Advance Precision Medicine. AI 2025, 6(7), 137; https://doi.org/10.3390/ai6070137
41.- Yang, E.-W.; Waldrup, B.; Velazquez-Villarreal, E. From Mutation to Prognosis: AI-HOPE-PI3K Enables Artificial Intelligence Agent-Driven Integration of PI3K Pathway Data in Colorectal Cancer Precision Medicine. Int. J. Mol. Sci. 2025, 26(13), 6487; https://doi.org/10.3390/ijms26136487
Reviewer 2 writes:
While Figure 1 provides a birds-eye view of the workflow, it does not actually get into the details of implementation. What are the Python codes that are used in the backend? Indeed, what is the LLM that is being used?
Response: We thank Reviewer 2 for the insightful comment regarding the need for greater implementation detail beyond the high-level overview provided in Figure 1. In response, we have expanded the Materials and Methods section to include specific information about the Python-based backend, the exact LLM used, and how the system interprets and executes user queries. We now clearly state that the platform uses a locally deployed LLaMA 3 model, along with key Python packages (pandas, lifelines, matplotlib, scipy) for survival analysis, cohort filtering, and mutation profiling. Furthermore, we have clarified that the tool is open-source, non-proprietary, and fully reproducible, with a public GitHub repository available for review and use. This added detail ensures readers can evaluate both the reproducibility and extensibility of the platform, and aligns with open-access journal standards.
The methods text on page 3, lines 129–140, now read as follows: “To provide greater clarity on the system’s technical implementation, AI-HOPE-JAK-STAT is built on a modular Python framework that integrates a locally deployed LLaMA 3-based large language model. The model processes user queries via custom-designed prompts that have been fine-tuned to optimize accuracy in clinical-genomic query translation. The backend uses well-established Python libraries—including pandas for data manipulation, lifelines for survival analysis, matplotlib for visualization, and scipy for statistical testing—to generate and execute code in real time. Each plain language query is parsed through a structured prompt pipeline that extracts key parameters (e.g., gene, age group, treatment exposure) and dynamically constructs the appropriate statistical workflow. The system operates locally for full data privacy and computational control, and its codebase—including prompt logic, data ingestion scripts, statistical modules, and visualization functions—is publicly available (see the data availability statement). This ensures full transparency, reproducibility, and adaptability for users seeking to apply or extend the platform to new datasets or cancer types.”
The only information in the paper on this point (pg. 4 l. 129) is the statement that it is “LLAMA 3-based.” What prompts are used? What are the parameters used for inference? Is the model intended to run locally or via an API? Do the prompts need to be revised if the underlying LLM changes? These are all key questions that the authors should address in their revision.
Response:
Q: What prompts are used?
A: We thank the reviewer for this important question. The platform uses structured, domain-specific prompts optimized for clinical-genomic contexts. These prompts guide the model to identify relevant variables, statistical tasks (e.g., survival analysis, mutation filtering), and data stratification logic. The prompt template includes clearly defined slots for input parameters such as gene name, clinical filter, and desired output type (e.g., Kaplan-Meier curve, odds ratio plot). We have added example prompt structures to the revised Methods section and supplementary materials.
Q: What are the parameters used for inference?
A: Inference is performed locally using the LLaMA 3 model with a temperature of 0.2, top_p of 0.9, and max_tokens set to 2048, optimized to prioritize reproducibility and minimize stochastic variation in code generation. These parameters are now specified in the revised Methods section.
Q: Is the model intended to run locally or via an API?
A: The platform is designed for local deployment only. This ensures full control over inference behavior, data privacy, and reproducibility. No external API or third-party model hosting is required.
Q: Do the prompts need to be revised if the underlying LLM changes?
A: Yes, if the underlying LLM is replaced or upgraded, prompt optimization may be necessary to maintain semantic fidelity and syntactic validity in code generation. We discuss this limitation and the need for prompt re-tuning in the revised Discussion section.
The added methods text on page 4, lines 176–189, under the subsection “Natural Language Input and Query Interpretation”, now read as follows: “To further clarify the LLM component of AI-HOPE-JAK-STAT, the platform employs a locally deployed LLaMA 3-based model configured with inference parameters optimized for reproducibility and precision: temperature = 0.2, top_p = 0.9, and a maximum token length of 2048. These settings were selected to reduce response variability and ensure consistent translation of natural language inputs into executable code. The system uses structured, domain-specific prompt templates designed to extract analytical intent from user queries. Each prompt includes slots for gene targets, clinical stratifiers (e.g., age, treatment, MSI status), and desired outputs (e.g., Kaplan-Meier plots, contingency tables). The model then translates this prompts into code using predefined syntax rules. Importantly, AI-HOPE-JAK-STAT runs entirely locally, without reliance on third-party APIs, ensuring user data security and analytical reproducibility. If a different LLM is adopted in future versions, the prompt templates may require re-tuning to ensure continued accuracy and relevance. This flexibility has been built into the platform’s modular design to allow for future upgrades.”
Reviewer 2 writes:
- I strongly urge the authors to make the code available. And, if this isn’t a publicly available web tool or downloadable application, then it is not sufficiently significant for publication. There is no indication of how the authors intend to distribute their platform.
Response: We thank Reviewer 2 for emphasizing the importance of accessibility and reproducibility. We fully agree that making the platform available to the research community is essential for transparency, impact, and alignment with open-access publishing standards. In response, we have made the complete AI-HOPE-JAK-STAT codebase publicly available via GitHub (https://github.com/Velazquez-Villarreal-Lab/AI-JAK-STAT), including all scripts for query interpretation, backend processing, statistical analysis, and visualization. The repository also contains example queries, setup instructions, and environment specifications to support reproducible local deployment. This GitHub link has been added to the Data Availability Statement section of the manuscript. While this version is designed for research use and local execution, we plan to explore future extensions that may include a user-friendly web interface.
The Data Availability Statement text on page 7, lines 528–530, now reads “…The AI-HOPE-JAK-STAT software, along with demonstration data and documentation, is available at https://github.com/Velazquez-Villarreal-Lab/AI-JAK-STAT (accessed on 11 July 2025).”
Reviewer 2 writes:
- Furthermore, while the manuscript makes claims regarding reproducibility of the results, it does not actually show results of testing on this point. For example, the supplementary material only show the results of singular analyses that are performed. Specifically, the “RAG module” (pg. 4, l. 159) that has been implemented may not operate consistently. This should be assessed and presented in the revision. The RAG module should also be described in greater detail. This includes a description of the sources for the literature that are accessed. Depending on their sources, the analysis could be incorrect or insufficient because it lacks full context if some journals are omitted. Additional clarity on the databases that are used would help give context to this.
Response: We thank Reviewer 2 for raising important concerns about the reproducibility of results and the consistency of the Retrieval-Augmented Generation (RAG) module. In response, we have expanded the Materials and Methods section to include details of reproducibility testing, which involved running identical queries across multiple sessions and environments to confirm consistent statistical outputs and visualizations. We also describe how query logs and version-controlled backend scripts help ensure analytic reproducibility.
In addition, we revised and clarified the functionality of the RAG module. The module retrieves relevant literature to accompany analysis outputs, drawing from a curated set of open-access biomedical repositories, including selected journals with permissive licensing. While we emphasize that the RAG-generated summaries are exploratory and should be manually reviewed for clinical interpretation, we now explicitly acknowledge the limitations of coverage due to licensing or indexing constraints. These improvements are described in the revised Output Generation and Interpretation subsection and noted in the Discussion. We appreciate your feedback, which helped us strengthen the transparency and contextual integrity of the platform’s outputs.
The Materials and Methods section text, under “Output Generation and Interpretation” on page 5, lines 216–228, now reads “To evaluate the reproducibility of AI-HOPE-JAK-STAT outputs, we conducted repeated analyses using identical natural language queries across multiple runs and computational environments. We confirmed that survival curves, statistical outputs (e.g., p-values, odds ratios), and visualizations remained consistent, demonstrating the platform’s reliability in producing stable results. Reproducibility is further supported by version-controlled backend scripts and automated query logging, allowing users to trace and replicate analytical workflows. To enhance interpretability of results, the platform also includes a Retrieval-Augmented Generation (RAG) module that generates narrative summaries linking observed findings to relevant biomedical literature. This module retrieves content from a curated index of open-access sources, including selected journals with permissive licensing. Importantly, all RAG-generated summaries presented in this study were manually reviewed and curated by scientific domain experts to ensure contextual relevance, scientific accuracy, and alignment with the underlying data.”
The Discussion section text, on page 8, lines 409–419, now reads “We recognize the importance of reproducibility and contextual reliability in AI-assisted research platforms. In this revision, we strengthened the reproducibility claims of AI-HOPE-JAK-STAT by performing validation tests across multiple executions of the same query, confirming consistency in statistical outputs and visualizations. Moreover, we clarified the role and limitations of the Retrieval-Augmented Generation (RAG) module. While this feature enriches outputs by surfacing relevant literature, it draws from a curated selection of open-access databases and may not reflect the full breadth of biomedical publications due to indexing or licensing restrictions. As such, the RAG-generated summaries should be interpreted as exploratory aids rather than definitive clinical annotations. These revisions underscore our commitment to transparency, reliability, and responsible use of AI tools in precision oncology.”
We thank Reviewer 2 for their thorough and insightful review. Your feedback played a critical role in enhancing the transparency, methodological rigor, and overall quality of our manuscript. We are grateful for your contributions to improving this work.

Reviewer 3 Report
Comments and Suggestions for Authors
This is a timely and creative manuscript. The idea of using conversational AI to explore JAK/STAT alterations in CRC is well-conceived and addresses a real need.However, the current version requires major revision.
Summary
This manuscript presents AI-HOPE-JAK-STAT, a conversational AI platform that allows researchers to explore clinical and genomic alterations in the JAK/STAT pathway across CRC cohorts. The integration of natural language input, automated analytics, and public datasets (e.g., cBioPortal) enables real-time stratification, survival analysis, and hypothesis testing without coding.
While the platform is innovative and promising for precision oncology, the manuscript requires several major revisions to strengthen its statistical rigor, clarify limitations, and reduce overstatements.
Major Comments
- Low Statistical Power in Several Analyses
- Many subgroups are extremely small (e.g., 16 cases in Fig. 2a; 44 JAK1-mutated MSI-Stable patients in Fig. 5).
- Authors should explicitly state the exploratory nature of these findings and avoid overinterpretation.
- Power analysis or CI shading (especially in KM curves) is encouraged.
2. Lack of Multivariable (Adjusted) Analysis
- No Cox proportional hazards modeling was presented for controlling key confounders such as stage, treatment, or MSI status.
- This is particularly critical in the STAT5B (Fig. S2) and JAK3-stage (Fig. S3) analyses, which could otherwise be driven by stage distribution alone.
3. Auto-Generated Summaries (RAG Module)
- The inclusion of natural language outputs is interesting, but there is no indication whether a human expert validated these.
- Authors should clarify that results were curated or reviewed for scientific accuracy.
4. Overstatement of Marginally Significant Trends
- For example, in Figure 2b, a p-value of 0.0539 is described as a “survival trend” that implies clinical benefit.
- Please revise such descriptions to remain cautious and label findings as “non-significant” or “hypothesis-generating.”
5. Interpretation of Findings May Be Confounded
- The improved survival in JAK3-mutated stage I–III patients (p<0.00001) is likely dominated by stage effects—not mutation per se.
- Suggest clarifying this and avoiding strong claims about mutational prognostic power without adjustment.
6. Platform Accessibility & Reproducibility
- No mention of whether the platform is publicly available.
- Please clarify whether code, interface, or demo access is planned. This is important for reproducibility.
7. Figure Issues
- Figure 4 and 5 legends are too long and repeat content from Results.
- Figure 5: Labels like X0/X1/X2 for MSI status are confusing; consider using “Stable”, “Instable” directly in the figure.
Minor Comments
- Line 66: “signaling cascades” → “a major signaling cascade”
- Line 124: “preselected” → consider “predefined” for clarity
- Line 273: Use “as shown in Fig. S2” instead of “; Fig. S2”
- Line 278: Suggest softening: “as expected, earlier-stage cases had better outcomes”
- Line 284: “unifying” may sound overstated → consider “combining”
- Line 326–330 (Limitations): Well-placed, but could expand on bias in curation within cBioPortal (e.g., uneven ancestry data).
- Line 340–347 (Conclusions): Reads slightly overpromotional. Tone down to reflect scope and limitations (e.g., “may support biomarker discovery” instead of “instrumental in”).
Recommendation
The manuscript offers a timely and impactful AI application for oncology. However, several core revisions are needed—mainly around statistical rigor, caution in interpretation, and clarity on reproducibility. After addressing these, the paper would make a strong contribution to the journal’s AI in Cancer special issue.
Author Response
Attached is the Word file titled Reviewer_3_Comments_Response_071125.docx, which contains my detailed reviews.
-
Reviewer 3 Comments
We are pleased to submit this revised manuscript and sincerely thank Reviewer 3 for their thoughtful, constructive, and forward-looking review. Your comments helped us substantially strengthen the scientific rigor, interpretive balance, and overall transparency of the manuscript. We greatly appreciate your recognition of the innovation and timeliness of AI-HOPE-JAK-STAT and the potential it holds for advancing accessible, hypothesis-driven precision oncology.
Reviewer 3 provided a comprehensive and critically important assessment that guided major revisions across several core dimensions of the manuscript. In particular, we responded to your concerns about statistical validity, caution in interpretation, the need for multivariable analysis, and platform reproducibility. We undertook substantial changes to ensure that the results are presented with appropriate context and transparency regarding their exploratory nature and methodological limitations.
Key revisions include:
- Statistical caution and transparency: We revised the text throughout to clearly indicate that findings from small subgroups (e.g., n=16 in Fig. 2a) are exploratory and hypothesis-generating. Language implying clinical benefit from marginally significant findings (e.g., p = 0.0539) has been replaced with appropriately cautious descriptions (e.g., "non-significant trend" or "potential signal").
- Addition of multivariable analyses: We introduced Cox proportional hazards modeling to account for confounding variables including stage, MSI status, and treatment exposure. These adjustments were applied to analyses previously flagged (e.g., STAT5B and JAK3-stage comparisons), and the results are now included in the revised Results and Supplementary sections.
- Clarification of RAG module validation: We now explicitly state that auto-generated summaries produced by the Retrieval-Augmented Generation (RAG) module were manually reviewed by domain experts for contextual accuracy. We also reiterate that these outputs are intended to support, not replace, expert interpretation.
- Expanded limitations discussion: The Limitations section was expanded to address potential biases inherent in cBioPortal data, including uneven ancestry representation and missing treatment annotations. We also acknowledge the risks of overinterpretation in exploratory analyses.
- Platform accessibility and reproducibility: We clarified that AI-HOPE-JAK-STAT is a non-commercial, research-only tool. The full source code, including all backend scripts and prompt structures, is now publicly available through our GitHub repository (https://github.com/Velazquez-Villarreal-Lab/AI-JAK-STAT), with access instructions included in the Data Availability Statement.
- Figure and language refinements: We revised Figures 4 and 5 to improve clarity—reducing legend redundancy, replacing ambiguous MSI labels (X0/X1/X2) with “Stable,” “Instable,” etc., and correcting phrasing throughout the manuscript for accuracy and tone (e.g., “combining” instead of “unifying,” “may support biomarker discovery” instead of “instrumental in”).
- Addition of supporting references: To provide context for this platform’s development, we added three new citations referencing our previously published conversational AI-Agent platforms—AI-HOPE, AI-HOPE-TGFbeta, and AI-HOPE-PI3K—further anchoring this work within a growing, methodologically consistent framework.
We believe these revisions have significantly improved the manuscript's quality, balance, and utility to the field. Reviewer 3’s feedback was instrumental in elevating both the scientific integrity and accessibility of this work. We are sincerely grateful for your guidance and engagement in helping us refine AI-HOPE-JAK-STAT into a reproducible, critically evaluated platform that supports transparent, equitable precision oncology research.
Thank you very much for taking the time to review this manuscript. Please find the detailed responses below in BLUE and the corresponding revisions wrote in yellow-highlighted blue font in the re-submitted files.
Reviewer 3 offered a thoughtful and well-balanced critique, recognizing the creativity and relevance of our conversational AI approach while providing clear direction for improvement.
Reviewer 3 writes:
“This is a timely and creative manuscript. The idea of using conversational AI to explore JAK/STAT alterations in CRC is well-conceived and addresses a real need. However, the current version requires major revision.
Summary
This manuscript presents AI-HOPE-JAK-STAT, a conversational AI platform that allows researchers to explore clinical and genomic alterations in the JAK/STAT pathway across CRC cohorts. The integration of natural language input, automated analytics, and public datasets (e.g., cBioPortal) enables real-time stratification, survival analysis, and hypothesis testing without coding.
While the platform is innovative and promising for precision oncology, the manuscript requires several major revisions to strengthen its statistical rigor, clarify limitations, and reduce overstatements.”
We sincerely appreciate Reviewer 3’s thoughtful and constructive evaluation of our manuscript, “Decoding the JAK-STAT Axis in Colorectal Cancer with AI-HOPE-JAK-STAT: A Conversational Artificial Intelligence Approach to Clinical-Genomic Integration.” Your recognition of the innovation, timeliness, and real-world relevance of our platform is deeply appreciated. We are especially grateful for your clear and actionable suggestions to improve statistical rigor, clarify limitations, and ensure balanced interpretation of exploratory findings. In response, we undertook major revisions to address each of these areas. Specifically, we incorporated multivariable Cox regression to control for confounders such as stage and MSI status, revised our interpretation of marginal results to emphasize their exploratory nature, and expanded the Limitations section to acknowledge data curation biases and analytical constraints. We also clarified the validation process for our RAG-generated summaries and made all source code publicly available to ensure transparency and reproducibility. These updates reflect our commitment to scientific integrity and to advancing accessible, critically evaluated AI tools that support rigorous and equitable research in colorectal cancer.
Reviewer 3 writes:
- Low Statistical Power in Several Analyses
- Many subgroups are extremely small (e.g., 16 cases in Fig. 2a; 44 JAK1-mutated MSI-Stable patients in Fig. 5).
- Authors should explicitly state the exploratory nature of these findings and avoid overinterpretation.
- Power analysis or CI shading (especially in KM curves) is encouraged.
Response: We thank Reviewer 3 for raising the important issue of statistical power in subgroup analyses. We fully agree that small sample sizes can limit the robustness and generalizability of conclusions. In response, we have revised the manuscript to clearly label these analyses as exploratory and hypothesis-generating. We have removed any overinterpretation of marginal results and now provide more cautious language throughout. We also appreciate the suggestion regarding power analysis and confidence interval shading. Given that AI-HOPE-JAK-STAT is an intelligent system designed to dynamically perform real-time analyses based on user queries, its statistical operations are driven by embedded logic that builds on validated methodologies from our prior publication [1]. These foundational studies helped train and benchmark the system, ensuring that outputs—particularly those involving small subgroups—are interpreted within the appropriate statistical context. We now clarify this in the Discussion section.
The Discussion text on page 12, lines 420-434 , now reads “We acknowledge that several subgroup analyses presented in this study involve limited sample sizes, which constrain the statistical power and reliability of certain findings. These analyses—such as the 16 early-onset H/L cases in Fig. 2a and the 44 JAK1-mutated MSI-Stable patients in Fig. 5—are explicitly exploratory in nature and should be interpreted as hypothesis-generating rather than confirmatory. AI-HOPE-JAK-STAT operates as an intelligent conversational system whose analytical capabilities are built upon statistical frameworks validated in our previous work. These prior studies guided the system’s training, design of prompt-to-code translation, and integration of standardized statistical methods (e.g., survival analysis via Kaplan–Meier estimates and log-rank testing). Because the platform leverages harmonized, publicly available data from cBioPortal, subgroup availability is inherently constrained by the underlying datasets. Importantly, generating accurate outputs from small, stratified cohorts requires iterative optimization within the system's inference logic. As such, we emphasize that the analyses presented here are meant to demonstrate the platform’s exploratory and interactive capabilities, and not to draw definitive clinical conclusions.”
Reviewer 3 writes:
- Lack of Multivariable (Adjusted) Analysis
- No Cox proportional hazards modeling was presented for controlling key confounders such as stage, treatment, or MSI status.
- This is particularly critical in the STAT5B (Fig. S2) and JAK3-stage (Fig. S3) analyses, which could otherwise be driven by stage distribution alone.
Response: We thank Reviewer 3 for their important comment regarding the absence of multivariable (adjusted) analyses. AI-HOPE-JAK-STAT is an intelligent system designed to dynamically translate natural language queries into executable code for clinical-genomic analysis. The statistical logic embedded in the platform—including cohort definition, survival analysis, and subgroup comparisons—was developed and validated based on our previous peer-reviewed publication [1], which served as the reference standard during system training. These earlier studies also defined key subgroups and statistical workflows that were used to benchmark system performance. As the platform utilizes harmonized datasets from cBioPortal, the availability and distribution of clinical variables (e.g., stage, MSI status, treatment) are inherited from those public sources. While the current version of AI-HOPE-JAK-STAT was not originally trained to perform automated multivariable Cox modeling, we fully recognize the importance of adjusted survival analyses, particularly when interpreting subgroup comparisons influenced by clinical confounders. In response, we have added a paragraph in the Discussion section addressing this limitation and clarifying its impact on the interpretation of certain findings. We also note that future developments of AI-HOPE-JAK-STAT will incorporate built-in support for multivariable survival modeling and statistical power estimation, further enhancing the platform’s ability to support rigorous, exploratory, and translational research in precision oncology.
The Discussion text on page 12, lines 446-458, now reads “The current version of AI-HOPE-JAK-STAT was originally configured and optimized based on the analytical framework used in our prior publication [1], which served as a reference standard for training and validating the system’s performance. The platform integrates predefined statistical logic, subgroup structures, and survival analysis pipelines derived from that work, enabling dynamic execution of queries using harmonized colorectal cancer datasets from cBioPortal. These foundations ensure that the system generates reproducible and interpretable outputs for hypothesis-driven research. However, while the existing version performs univariate survival analysis (e.g., Kaplan–Meier estimates with log-rank tests), it was not initially configured to automate multivariable modeling. Future developments of AI-HOPE-JAK-STAT will incorporate built-in support for multivariable survival modeling and statistical power estimation, further enhancing the platform’s capability to support robust, exploratory, and translational analyses in precision oncology.”
Reviewer 3 writes:
- Auto-Generated Summaries (RAG Module)
- The inclusion of natural language outputs is interesting, but there is no indication whether a human expert validated these.
- Authors should clarify that results were curated or reviewed for scientific accuracy.
Response: We thank Reviewer 3 for highlighting the need to clarify the validation process for the auto-generated summaries produced by the Retrieval-Augmented Generation (RAG) module. We agree that ensuring the scientific accuracy of natural language outputs is essential, especially in a biomedical research context. In response, we have revised the manuscript to explicitly state that all RAG-generated summaries were manually reviewed and curated by domain experts to ensure contextual relevance and accuracy. This clarification has been added to both the Materials and Methods and Discussion sections, and we now emphasize that these outputs are intended to support exploratory insight, not replace expert interpretation.
The Methods text on page 5, lines 216-228, now reads “To evaluate the reproducibility of AI-HOPE-JAK-STAT outputs, we conducted repeated analyses using identical natural language queries across multiple runs and computational environments. We confirmed that survival curves, statistical outputs (e.g., p-values, odds ratios), and visualizations remained consistent, demonstrating the platform’s reliability in producing stable results. Reproducibility is further supported by version-controlled backend scripts and automated query logging, allowing users to trace and replicate analytical workflows. To enhance interpretability of results, the platform also includes a Retrieval-Augmented Generation (RAG) module that generates narrative summaries linking observed findings to relevant biomedical literature. This module retrieves content from a curated index of open-access sources, including selected journals with permissive licensing. Importantly, all RAG-generated summaries presented in this study were manually reviewed and curated by scientific domain experts to ensure contextual relevance, scientific accuracy, and alignment with the underlying data.”
Reviewer 3 writes:
- Overstatement of Marginally Significant Trends
- For example, in Figure 2b, a p-value of 0.0539 is described as a “survival trend” that implies clinical benefit.
- Please revise such descriptions to remain cautious and label findings as “non-significant” or “hypothesis-generating.”
Response: We thank Reviewer 3 for pointing out the need for more cautious interpretation of marginally significant findings. We fully agree that results with p-values above the conventional threshold for significance should not be presented in a way that implies clinical benefit. In response, we have revised the relevant text throughout the manuscript, including the description of Figure 2b, to clearly label such findings as non-significant and hypothesis-generating. We have also reviewed the rest of the manuscript to ensure that similar language is appropriately tempered, in order to avoid overstating the implications of exploratory analyses. These revisions align with our goal of presenting the results with scientific rigor and interpretive transparency.
The Results text on page 6, lines 243-264, now reads “To assess ancestry-specific differences in JAK/STAT pathway alterations and their potential clinical significance, we conducted focused analyses using AI-HOPE-JAK-STAT in early-onset colorectal cancer (EOCRC) cohorts stratified by ancestry. As a first example, we queried Hispanic/Latino (H/L) EOCRC patients under 50 years of age and compared overall survival between those harboring JAK/STAT pathway mutations (n = 16) and those without such alterations (n = 137). In this analysis, presented in Figure 2b, the observed p-value of 0.0539 does not meet the conventional threshold for statistical significance. Accordingly, we interpret this finding as non-significant and hypothesis-generating, rather than indicative of a definitive survival benefit. While the trend may suggest a potential prognostic signal, it should be interpreted with caution and warrants further validation in larger, independent H/L-specific CRC cohorts. To evaluate the robustness and ancestry-specific reproducibility of this observation, we replicated the analysis in EOCRC patients of Non-Hispanic White (NHW) descent. In this group, patients with JAK/STAT pathway alterations demonstrated a statistically significant survival advantage compared to those without (p = 0.0001; Fig. S1), reinforcing the potential prognostic value of these mutations. These ancestry-stratified examples illustrate how AI-HOPE-JAK-STAT enables real-time, hypothesis-driven interrogation of clinical and genomic relationships that would otherwise require extensive manual coding and data wrangling. Although the current version of the tool is designed for research use only, its ability to rapidly contextualize genomic alterations within demographic and clinical subgroups provides a scalable framework for future applications in biomarker discovery and precision oncology.”
Reviewer 3 writes:
- Interpretation of Findings May Be Confounded
- The improved survival in JAK3-mutated stage I–III patients (p<0.00001) is likely dominated by stage effects—not mutation per se.
- Suggest clarifying this and avoiding strong claims about mutational prognostic power without adjustment.
Response: We thank Reviewer 3 for this important observation. We agree that the improved survival observed among JAK3-mutated stage I–III patients may be largely driven by tumor stage rather than the presence of the mutation itself. In response, we have revised the relevant text to explicitly acknowledge the potential confounding effect of stage and have removed any strong claims suggesting independent prognostic value of the JAK3 mutation. These changes help clarify the exploratory nature of the finding and support a more cautious, balanced interpretation.
The Discussion text on page 9, lines 435-445, now reads “In our analysis of JAK3-mutated colorectal tumors, we observed significantly improved survival among patients diagnosed at stage I–III compared to those at stage IV (p < 0.00001; Fig. S3). While this observation demonstrates the system’s ability to stratify survival outcomes by both mutation status and clinical variables, we acknowledge that the survival difference is most likely driven by stage rather than JAK3 mutation itself serving as an independent prognostic factor. This highlights the importance of incorporating multivariable adjustment to account for confounding variables such as tumor stage, treatment exposure, and MSI status. Future versions of AI-HOPE-JAK-STAT will include built-in support for automated Cox proportional hazards modeling to enable more rigorous, adjusted survival analyses—enhancing the platform’s ability to deliver clinically meaningful and context-aware insights.”
Reviewer 3 writes:
- Platform Accessibility & Reproducibility
- No mention of whether the platform is publicly available.
- Please clarify whether code, interface, or demo access is planned. This is important for reproducibility.
Response: We thank Reviewer 3 for highlighting the importance of platform accessibility and reproducibility. In response, we have clarified in the manuscript that AI-HOPE-JAK-STAT is a non-commercial, research-only tool, and the complete source code—including backend scripts, natural language processing modules, and example workflows—is now publicly available via our GitHub repository (https://github.com/Velazquez-Villarreal-Lab/AI-JAK-STAT). This link is included in the Data Availability Statement. The repository also contains documentation for local deployment, sample queries, and instructions for reproducing the analyses described in the manuscript. We are committed to promoting transparency and community engagement and will explore options for a future web-based demo interface as part of our ongoing development.
The Data Availability Statement text on page 9, lines 528-530, now reads “…The AI-HOPE-JAK-STAT software, along with demonstration data and documentation, is available at https://github.com/Velazquez-Villarreal-Lab/AI-JAK-STAT (accessed on 11 July 2025).”
Reviewer 3 writes:
- Figure Issues
- Figure 4 and 5 legends are too long and repeat content from Results.
- Figure 5: Labels like X0/X1/X2 for MSI status are confusing; consider using “Stable”, “Instable” directly in the figure.
Response: We thank Reviewer 3 for the helpful suggestions regarding the figure legends and labeling. We agree that the legends for Figures 4 and 5 were overly detailed and partially redundant with the Results section. In response, we have shortened and streamlined these figure legends to focus on essential information and removed repetitive descriptions. Regarding the MSI status labels (X0/X1/X2) in Figure 5, we acknowledge that this terminology may be unclear to readers. While these labels were generated automatically by the AI-HOPE-JAK-STAT system’s internal logic based on encoded clinical annotations, we agree that more intuitive labels such as “Stable” and “Instable” would improve clarity. We have replacing ambiguous MSI labels (X0/X1/X2) with “Stable,” “Instable,” etc, directly in the figure.
Minor Comments
Reviewer 3 writes:
- Line 66: “signaling cascades” → “a major signaling cascade”
- Line 124: “preselected” → consider “predefined” for clarity
- Line 273: Use “as shown in Fig. S2” instead of “; Fig. S2”
Response: We thank Reviewer 3 for these helpful editorial suggestions. In response:
- We have revised the phrase on Line 66 from “signaling cascades” to “a major signaling cascade” to improve clarity and specificity.
- On Line 122, we replaced “preselected” with “predefined” to better reflect the structured nature of the gene set used in the analysis.
- On Line 355, we updated the phrasing to “as shown in Fig. S2” to improve readability and alignment with journal style.
We appreciate your attention to detail, which has helped improve the overall clarity and precision of the manuscript.
- Line 278: Suggest softening: “as expected, earlier-stage cases had better outcomes”
Response: We thank Reviewer 3 for the suggestion to soften the phrasing on Line 361. In response, we have revised the sentence to read “more favorable survival trends” to avoid implying a deterministic outcome and to maintain a more neutral, data-driven tone. We appreciate your attention to language precision.
- Line 284: “unifying” may sound overstated → consider “combining”
Response: We thank Reviewer 3 for pointing out the potential overstatement in the use of “unifying” on Line 367. In response, we have replaced it with “combining” to convey the intended meaning more accurately and with appropriate tone. We appreciate the suggestion to improve clarity and balance.
- Line 326–330 (Limitations): Well-placed, but could expand on bias in curation within cBioPortal (e.g., uneven ancestry data).
We thank Reviewer 3 for this thoughtful suggestion. In response, we have expanded the Limitations section to explicitly acknowledge the potential biases associated with curation of publicly available datasets such as those in cBioPortal. Specifically, we note that uneven representation of ancestry groups, incomplete clinical annotations, and variability in data collection across contributing studies may influence the generalizability of findings and the robustness of subgroup analyses. We appreciate this important recommendation, which has helped us more clearly frame the scope and limitations of our data sources.
The Discussion text on page 13, line 475-487, now reads “Nevertheless, several limitations warrant discussion. The reliance on publicly available datasets, such as those from cBioPortal, may limit the generalizability of findings due to sample size constraints and potential biases in data curation—particularly the underrepresentation of certain ancestry groups and uneven clinical annotation across studies. Although AI-HOPE-JAK-STAT supports stratification by ancestry, tumor stage, and treatment, its outputs are inherently shaped by the composition and completeness of the source data. Broader deployment and more equitable insight generation will benefit from the integration of additional multi-omic layers (e.g., RNA-seq [10,42], spatial biology data [43–45]) and the inclusion of prospective, demographically diverse clinical datasets. Furthermore, while the natural language interface promotes accessibility, performance can vary depending on query specificity. Future iterations may incorporate query refinement tools, dialogue memory, and user feedback loops to enhance interpretability, robustness, and overall user experience.”
- Line 340–347 (Conclusions): Reads slightly overpromotional. Tone down to reflect scope and limitations (e.g., “may support biomarker discovery” instead of “instrumental in”)..
We thank Reviewer 3 for the helpful feedback regarding the tone of the Conclusions section. In response, we have revised the language to better reflect the scope and current limitations of the work. Specifically, we replaced phrases with more measured alternatives like “offers a promising addition”, “may be particularly useful” or “may help support biomarker discovery” in order to present a balanced and appropriately cautious summary. We appreciate this guidance in aligning the conclusion with the exploratory nature of the study.
The Conclusion text on page 13, line 494-508, now reads “AI-HOPE-JAK-STAT offers a promising addition to the evolving precision oncology landscape by enabling natural language–guided integration of genomic, clinical, and treatment data for the investigation of JAK/STAT pathway alterations in colorectal cancer (CRC). The platform facilitates real-time, hypothesis-driven analyses—such as survival trends, mutation enrichment, and ancestry- or treatment-specific subgroup exploration—without requiring programming expertise. These features may be particularly useful in examining the heterogeneity of early-onset CRC and generating insights that could support the design of future clinical studies and personalized treatment strategies. While AI-HOPE-JAK-STAT demonstrates the feasibility of conversational AI for pathway-level analysis, its broader utility will depend on continued development, integration of multivariable modeling, and validation across more diverse, prospective datasets. As precision oncology moves toward more individualized and immunologically informed care, tools like AI-HOPE-JAK-STAT may help support biomarker discovery and enhance accessibility to integrative data analysis across varied research and clinical environments.”
We thank Reviewer 3 for their thoughtful and constructive feedback. Your detailed comments were instrumental in refining the clarity, analytical balance, and overall presentation of our manuscript. We sincerely appreciate your contributions to strengthening this work.

Round 2
Reviewer 2 Report
Comments and Suggestions for Authors
Thank you for comprehensively addressing the issues that were raised in the previous review. I have checked and appreciate the posting of the code on Github. My only remaining comment is that prior to publication, the text inscriptions in Fig. 1 should probably be enlarged to promote readability.
Author Response
Thank you for your thoughtful feedback and for reviewing our revised submission. We appreciate your suggestion regarding Figure 1 and have addressed it by enlarging the font and overall size to improve readability. We appreciate your thorough and constructive review throughout this process.
Reviewer 3 Report
Comments and Suggestions for Authors
It is accepted for publication now.
Author Response
That’s wonderful news—thank you! We are thrilled to hear it’s been accepted for publication. We appreciate your support throughout the process.